

# Ecological and taxonomic dissimilarity in species and higher taxa of reptiles in western Mexico

Jaime Manuel Calderón-Patrón[1], Jorge Téllez-López[2], Eréndira Canales-Gómez[2] and Karen Elizabeth Peña-Joya[2]

[1] Laboratorio de Biodiversidad de la Escuela de Ciencias, Universidad Autónoma Benito Juárez, Oaxaca, Oaxaca, Mexico
[2] Laboratorio de Ecología, Paisaje y Sociedad, Centro Universitario de la Costa, Universidad de Guadalajara, Puerto Vallarta, Jalisco, Mexico

Corresponding author
Karen Elizabeth Peña-Joya,
karen.joya@academicos.udg.mx

## ABSTRACT

Reptiles are one of the most diverse groups of vertebrates in the world that are distributed in almost all ecosystems. Many of these studies have focused on exploring their diversity patterns across different environments; and recent studies on reptile alpha and beta diversity have incorporated a multifaceted approach into their analysis to have more comprehensive evaluations. This study presents an assessment of the taxonomic diversity of reptile patterns using methods that incorporate the assessment of higher taxa. Likewise, the taxonomic dissimilarities between reptile communities in the physiographic regions of the state of Jalisco were analyzed. Evaluations for the groups of snakes and lizards independently are presented. We use the taxonomic distinctiveness index that assesses the complexity of the taxonomic structure of communities through hierarchical classification above the species level to measure the relationships between taxa. The dissimilarity of the taxonomic structure in each community was also analyzed. Beta diversity partitions were performed to identify the contribution of turnover and the differences in richness. We determined that alpha diversity of species and higher taxa maintain different patterns, indicating that Jalisco presents regions with overrepresentation of reptile families and genera, as well as regions with an opposite trend. The representation of higher taxa is higher in the lizard group, although in terms of species richness snakes are the most prominent group. The turnover is the most important component at species and higher taxa, with similar values for lizards and snakes. The findings presented show that incorporating phylogenetic information about species through taxonomic relationships provides complementary information that species diversity *per se*, especially at the level of alpha diversity.

# INTRODUCTION

Reptiles are one of the most diverse groups of vertebrates in the world, containing 12,162 species distributed in almost all ecosystems on the planet (*Uetz, 2024*). Reptiles are

characterized by a wide variety of behavioral, ecological, and life history strategies that have given them a central role as model organisms for ecological and evolutionary studies (*Shine, 2005*). Many of these studies have focused on exploring their diversity patterns across different environments (*e.g.*, *de Cervantes-López & Morante-Filho, 2024*; *Ortiz-Medina, Peña-Peniche & Chablé-Santos, 2022*; *Supsup et al., 2020*) and different scales (*e.g.*, *García, Solano-Rodríguez & Flores-Villela, 2007*; *Núñez et al., 2022*; *Qian, 2009*). Particularly, these evaluations seek to understand the patterns of local diversity or alpha diversity. This attribute represents the richness of species and the evenness of the distribution of their abundances (*Moreno et al., 2018*). In reptiles, it has been shown that alpha diversity is associated with high structural complexity in their habitats (*Bateman & Merritt, 2020*) higher temperature conditions (*Núñez et al., 2022*) and water availability (*Chiacchio et al., 2020*).

In addition to this, recent studies on reptile alpha diversity have incorporated a multifaceted approach into their analysis, following current methodological proposals that seek more precise measurements to reach more solid conclusions (*Moreno et al., 2018*). Some of the facets of biodiversity that have been included in the study of reptiles are the taxonomic, functional and phylogenetic facets (*Berriozabal-Islas et al., 2017*; *Chiacchio et al., 2020*; *Hernández-Salinas et al., 2023*; *Peña-Joya et al., 2020*; *Ramm et al., 2018*; *Rosas-Espinoza et al., 2024*). Regarding the taxonomic facet, this has been evaluated considering only species richness, which is the simplest way to evaluate the diversity of a community (*Magurran, 1988*). However, indices such as the taxonomic distinctiveness proposed by *Clarke & Warwick (1998)* have also been used, where complexity in the taxonomic structure of communities is evaluated through hierarchical classification above the species level to measure relationships between taxa (*Pérez-Hernández, 2019*). The authors who have applied this index in reptile diversity assessments have determined that at the local level, there is a high taxonomic distinctiveness as a result of the high representation of genera and families (*Cruz-Elizalde et al., 2022*; *Peña-Joya et al., 2018*). These studies also show an incongruence between the values of species richness and taxonomic distinctiveness, that is, that reptile communities have a low number of species, but a high representation of higher taxa (*Peña-Joya et al., 2018*). However, *Hernández-Salinas et al. (2023)* show that reptile communities present both a high richness of species and higher taxa. The contrasting results of these assessments are particularly interesting, since differences in biodiversity facets may suggest opposing conservation priorities (*Cadotte & Tucker, 2018*; *Martín-Regalado et al., 2022*).

Although numerous studies have focused on the evaluation of alpha diversity, there are increasingly more works focused on the analysis of species turnover or beta diversity (*Lewthwaite, Debinski & Kerr, 2017*; *Si, Baselga & Ding, 2015*). Beta diversity evaluates the differentiation in species composition between two or more communities (*Bishop et al., 2015*). To understand the ecological processes that determine this differentiation, beta diversity can be partitioning into the components of turnover and differences in richness; where turnover refers to the replacement of some species by others due to environmental or spatial restrictions, and differences in richness occur when communities with fewer species are subsets of communities with greater richness (*Baselga, 2010*). In reptiles, species

turnover is the most important component of beta diversity and is associated with the limited dispersal capacity of these organisms, as well as their niche limitations (*Calderón-Patrón et al., 2016*). Likewise, the relationship between the beta diversity of reptiles with abiotic factors such as elevation, temperature (*Jins et al., 2021*; *McCain, 2010*; *Whiting & Fox, 2021*) and precipitation (*Soares & Brito, 2006*) has been determined.

As in alpha diversity, the multifaceted approach has been extended to beta diversity (*Li et al., 2023*; *Qian & Qian, 2023*; *Rosas-Espinoza et al., 2024*). Regarding the taxonomic facet, some studies have carried out analysis of differentiation between communities including higher taxa (*García de Jesús et al., 2016*), even this taxonomic beta diversity has been partitioned into its turnover components and differences in richness (*Bacaro, Ricotta & Mazzoleni, 2007*; *Calderón-Patrón & Moreno, 2019*). Concerning this approach, a high taxonomic beta diversity has been reported for reptiles, which is mainly explained by the turnover of higher taxa (*Calderón-Patrón et al., 2013*, *2016*).

Based on this background, we carried out an assessment of the taxonomic diversity patterns of reptiles with methods that incorporate the evaluation of higher taxa. Likewise, we analyzed the taxonomic dissimilarities between reptile communities in the physiographic regions of the state of Jalisco, determining the ecological processes that establish this differentiation. Regarding this, we propose three main hypotheses: (i) the α diversity of species and higher taxa of reptiles show different patterns; (ii) the β diversity of species and higher taxa is mainly caused by the turnover component; (iii) lizards and snakes have different diversity patterns within and between region.

## MATERIALS AND METHODS

### Study area

The western region of Mexico, where the state of Jalisco is located, is considered one of the largest and most complex regions in the country. Its topographic complexity, as well as the influence of the Nearctic and Neotropical biogeographic regions, contribute to the great variety of environments and its high biological diversity, which includes a significant number of endemic and restricted-distribution species (*Chávez-Ávila et al., 2015*). This state has a great variety of ecosystems from tropical to temperate environments. Jalisco has a contrasting relief that includes large mountain systems, volcanic plains, foothills, valleys and hills. As a result of this complexity, the state has a large number of endemic species for Mexico, occupying seventh place in diversity of amphibians and reptiles at the national level (*Ochoa-Ochoa & Flores-Villela, 2006*).

The state of Jalisco (80,208.29 km$^2$) is located in the western center of Mexico at the confluence zone between the Sierra Madre Occidental, Sierra Madre del Sur and the Trans-Mexican Volcanic Belt (*Valdivia-Ornelas, 2018*); this region features complex orography, with elevations between 0 and 4,600 m asl (*Jardel-Peláez et al., 2017*). The area has dry, tropical, and temperate climates, the latter predominating and the average temperature being between −3 °C and 22 °C; additionally, it has a high incidence of hydrometeorological phenomena due to its location in the intertropical zone (*Valero-Padilla, Rodríguez-Reynaga & Cruz-Angón, 2017*; Fig. 1). For this work, we use the reptile records reported by *Cruz-Sáenz et al. (2017)* in the seven physiographic regions of Jalisco,

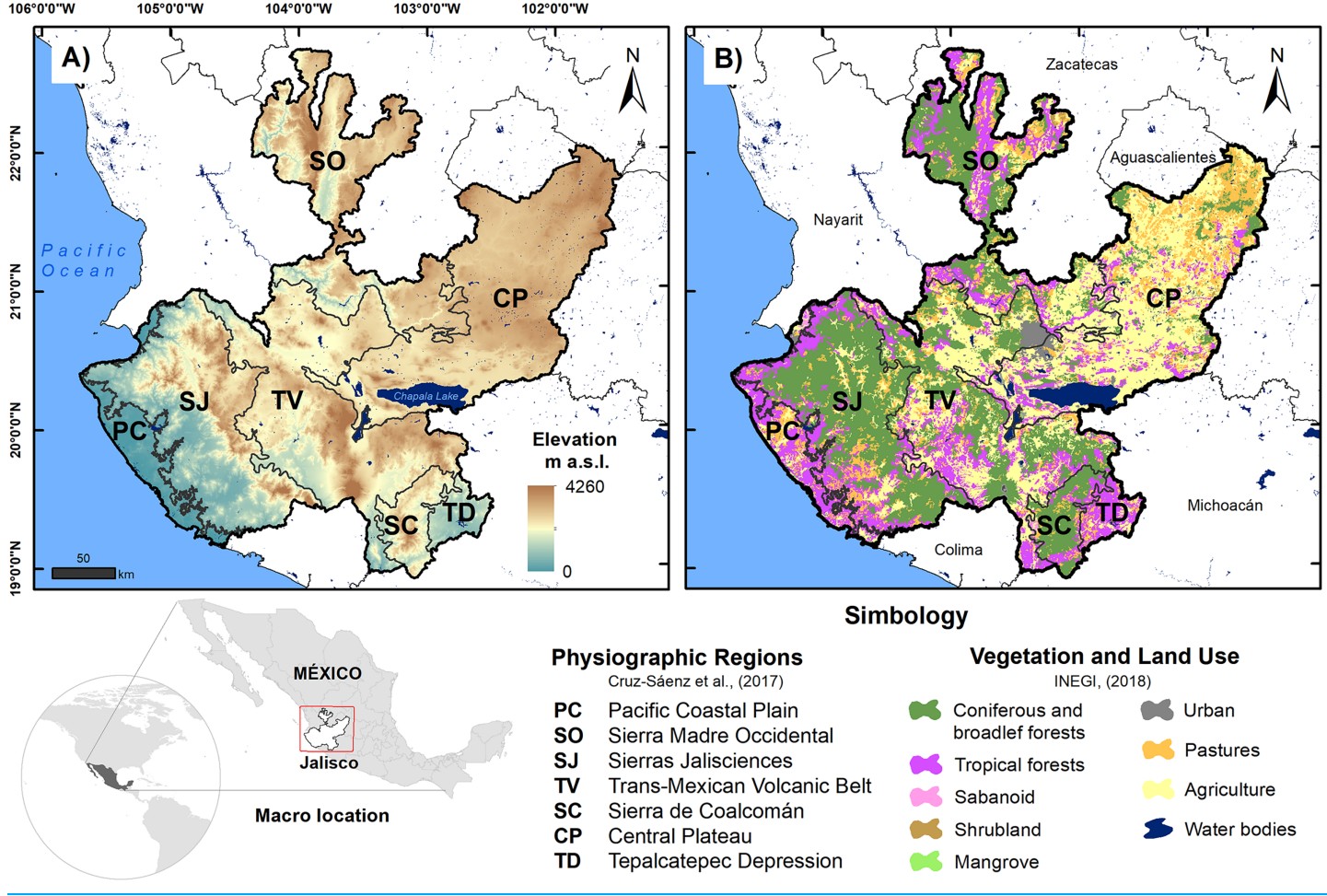

**Figure 1** Location map of physiographic regions of the state of Jalisco; (A) elevation is shown and (B) vegetation and land use are shown.
Source: *Cruz-Sáenz et al. (2017)* and *INEGI (2018)*.

which are the Pacific Coastal Plain (PC), Sierra Madre Occidental (SO), Sierras Jaliscienses (SJ), the Trans-Mexican Volcanic Belt (TV), Sierra de Coalcomán (SC), the Central Plateau (CP), and the Tepalcatepec Depression (TD). Using the records of the reptiles in these regions, we built a database of presence and absence. A summary of the main characteristics of each physiographic region is shown in Table 1.

## Data analysis

### Alpha diversity of reptiles at the species and higher taxa level

To determine the alpha taxonomic diversity of reptiles in physiographic regions, the average taxonomic distinctiveness index (AvTD) was used to evaluate the degree of taxonomic relationship between the species of an assemblage (*Clarke & Warwick, 1998*). For this purpose, an aggregation matrix was generated with six taxonomic levels (order, superfamily, family, genus subfamily and species), which was related to the presence and absence of the reptile species in the seven physiographic regions. The taxonomic levels of

**Table 1 Surface, elevation, annual average temperature, precipitation, and percentage of land use and land cover of the physiographic regions of Jalisco (Cruz-Sáenz et al., 2017).**

| Physiographic region | Area (km²)[1] | Elevation (m a.s.l.)[2] | | | Annual average temperature (°C)[3] | Annual precipitation (mm)[3] | Vegetation and land use (LU) (%)[4] | |
|---|---|---|---|---|---|---|---|---|
| | | Min | Mean | Max | | | Vegetation | LU |
| PC | 3,818.5 | 0 | 96.29 | 680 | 25.3 | 950.5 | 63.1%; TF (57.1%) | 36.9%; AGR (18.4%) |
| SO | 15,712.4 | 273 | 1,673.10 | 2,863 | 19.6 | 774.0 | 64.7%; CBF (36.1%) | 35.3%; AGR (21.2%) |
| SJ | 15,070.3 | 0 | 1,030.94 | 2,880 | 20.8 | 1,335.0 | 82.2%; CBF (57.2%) | 17.8%; PA (11.4%) |
| TV | 18,733.2 | 144 | 1,522.32 | 4,260 | 19.7 | 897.8 | 53.3%; CBF (33.2%) | 46.7%; AGR (33.2%) |
| SC | 2,714.9 | 484 | 1,367.62 | 2,725 | 21.0 | 938.8 | 86.1%; CBF (64.6%) | 13.9%; PA (11.2%) |
| CP | 20,702.8 | 1,252 | 1,839.61 | 2,957 | 18.2 | 716.7 | 24.3%; TF (10.4%) | 75.7%; AGR (53.4%) |
| TD | 1,551.9 | 327 | 715.57 | 1,704 | 25.1 | 843.8 | 70.7%; TF (68.2%) | 29.3%; AGR (15.5%) |

Note:
Physiographic regions: Pacific Coastal Plain (PC); Sierra Madre Occidental (SO); Sierras Jaliscienses (SJ); Trans-Mexican Volcanic Belt (TV); Sierra de Coalcomán (SC); Central Plateau (CP); Tepalcatepec Depression (TD). Source: [1]Cruz-Sáenz et al. (2017); [2]INEGI (2013); [3]Fick & Hijmans (2017); [4]INEGI (2018). Abbreviations Vegetation and Land Use. CBF, coniferous and broadleaf forests; TF, tropical forests; AGR, agriculture; PA, pastures.

the aggregation matrix were weighted according to the criteria established by Clarke & Gorley (2006). The average taxonomic distinctiveness and confidence intervals less than and greater than 95% were calculated based on 1,000 random interactions using the PRIMER V7® program (Clarke & Gorley, 2015). This analysis was also performed independently for groups of lizards and snakes following the same process and considering the same criteria.

*Partitions of reptile beta diversity at the species and higher taxa level*

The beta diversity analyses of the physiographic regions were performed independently for all reptiles, as well as for the group of lizards and snakes, as we believe that these groups have different environmental requirements and that their beta diversity patterns may change. For this purpose, the procedure of Carvalho, Cardoso & Gomes (2012) and Carvalho et al. (2013), which is based on the approach of Baselga (2010, 2012) and Baselga & Leprieur (2015), is used to separate beta diversity into two components. According to this method, the total dissimilarity (β.sor) is one minus the Sorensen coefficient of similarity. This total dissimilarity was divided into two components: dissimilarity due to turnover (β.sim) and dissimilarity due to differences in richness (βsne). In two hypothetical communities (1 and 2) b are the species exclusive to community 1 and c are the species exclusive to community 2 and a are the species shared between both communities. Total beta diversity: ßsor = b + c/a + b + c, turnover: βsim = 2 * min (b, c)/a + b + c and differences in richness: βsne = (b−c)/a + b + c. This analysis was carried out in R using the betapart package specifically the *beta.multi* and *beta.pair* functions (Baselga & Orme, 2012; R Core Team, 2018).

We also partitioned the dissimilarity in the taxonomic structure considering the composition of the higher taxa (order, superfamily, family, genus, subfamily, and species). For this case, according to the methods of Bacaro, Ricotta & Mazzoleni (2007), the total taxonomic dissimilarity, here β.sorT (1−ΔT *sensu* Bacaro, Ricotta & Mazzoleni, 2007), is
equal to the dissimilarity of the Sorensen coefficient but considers more taxa. Taxonomic dissimilarity was measured as $\beta sorT = 1-(Ta/Ta + Tb + Tc)$, where Ta is the total number of taxa shared between two communities, Tb is the number of taxa present only in the first community but absent in the second, and Tc is the number of taxa present exclusively in the second community. The values of β.sorT range from 0 when the taxonomic structure of both communities is identical to 1 when the taxonomic structure is completely different (*Bacaro, Ricotta & Mazzoleni, 2007*). The taxonomic data were calculated as the proportion of nonshared taxa relative to the total number of taxa in the two communities. Therefore, the partition of β.sorT with the procedure of *Carvalho et al. (2013)* shows a dissimilarity component due to the change in taxa (β.simT) and a compound number of dissimilarities due to the difference in the richness of taxa (β.sneT). The analysis was carried out in R (*R Core Team, 2018*) following *Carvalho et al. (2013)*.

### Dissimilarity of reptiles at the species and higher taxa level

To represent the species and taxonomic dissimilarity between the physiographic regions, cluster analyses were carried out, which were constructed by unweighted pair group method with arithmetic mean (UPGMA; *Clarke & Gorley, 2015*). Cluster analyses were performed independently for the reptile group and for the lizard and snake groups. The groups were described at 50% dissimilarity at the species level and 40% at the higher taxon level to interpret dissimilarity proportionally.

### Relationships between beta diversity of reptile at the species and higher taxa level

Finally, to determine if there is congruence between the beta diversity of species and higher taxa, non-parametric correlations were performed for total beta diversity (β. sor), turnover (β. sim) and differences in richness (β. sne) in the seven physiographic regions. This analysis was carried out considering the totality of the reptiles as well as the groups of lizards and snakes.

## RESULTS

### Alpha diversity of reptiles at the species and higher taxa level

The physiographic region with the highest reptile richness was TV with 85 species, followed by PC with 75 species (Fig. 2A). The regions with the lowest richness were the SC and TD regions, with 24 and 23 species, respectively. The remaining regions had a richness of reptiles between 63 and 68 species. The taxonomic distinctiveness followed a different pattern than the species richness; the PC region presented the highest distinctiveness value (64.81), and it was significantly higher than expected ($p \leq 0.05$). The rest of the regions maintained distinctiveness values between 52.97 and 56.18, except the TD region, which obtained the lowest distinctiveness value (49.15). Notably, the CP, SJ, and TV regions are completely outside the probability funnel, which indicates that their taxonomic distinctiveness is significantly lower than expected ($p \leq 0.05$).

For the lizard group, TV was the physiographic region with the highest richness, with 29 species, followed by CP, with 26 species (Fig. 2B). The regions with the lowest lizard

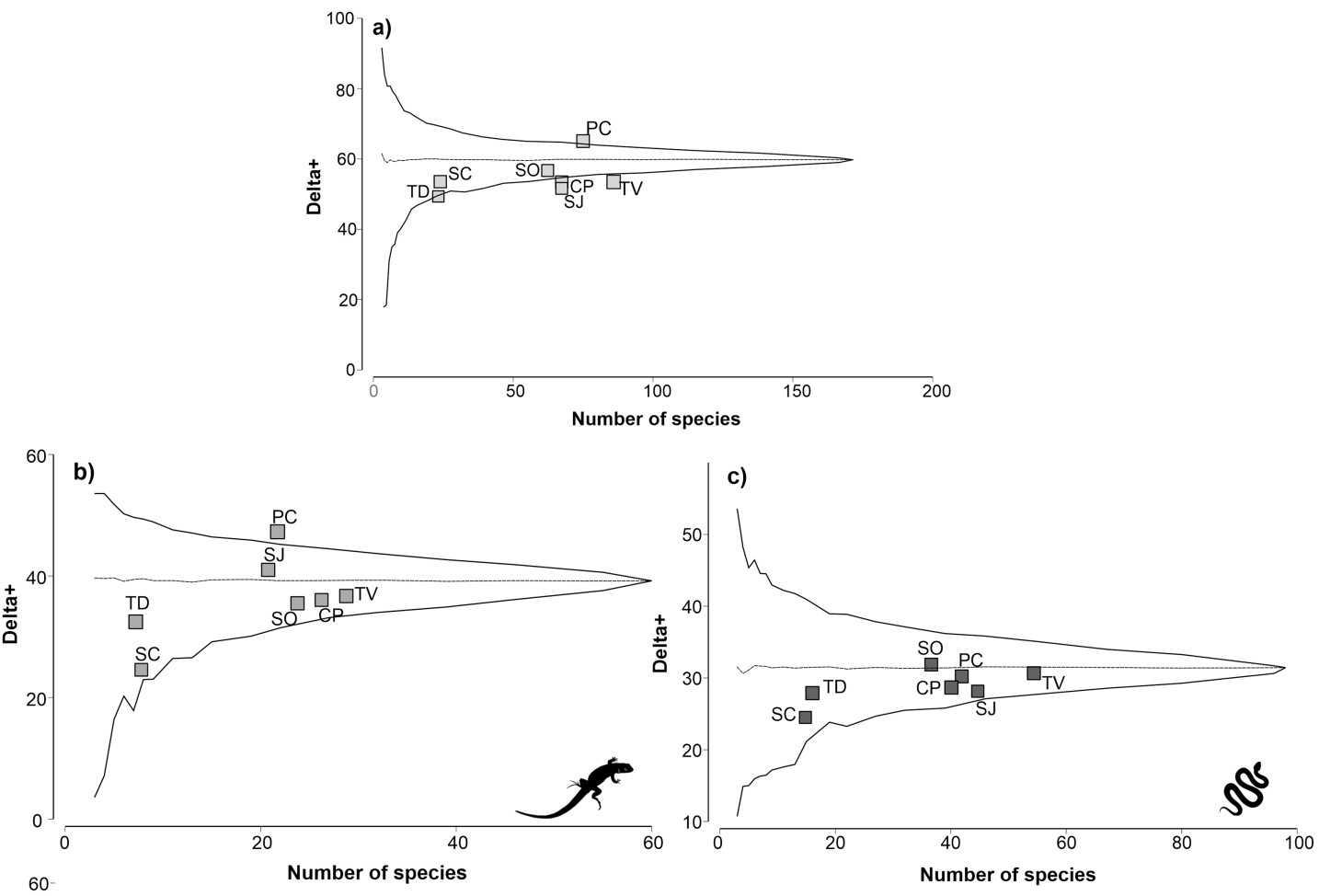

**Figure 2 Analysis of taxonomic distinctness of the of the physiographic regions for reptiles (A), lizards (B) and snakes (C).** The dotted line represents the average taxonomic distinctness; the solid lines represent the lower and upper confidence intervals at the 95% level. Lizard and snake silhouettes: https://images.app.goo.gl/wfvYcDTbBktp8mFZ7; snake cobra or anaconda silhouette icon long vector image (vectorstock.com).

richness were the SC and TD regions, with eight and seven species, respectively. Regarding taxonomic distinctiveness, the PC region presented the highest value (46.83), which was significantly greater than expected ($p \leq 0.05$). The rest of the regions had values between 32.65 and 40.63, with the TD region having the least taxonomic distinctiveness. For the lizard group, all regions except the PC region remained within the probability funnel, indicating that the taxonomic distinctiveness of the regions was consistent with expectations ($p > 0.05$).

For the snakes group, the TV region with the highest species richness (54) was also highlighted, followed by the SJ region (45) (Fig. 2C). The regions with the lowest richness of snakes were SC and TD, with 15 and 16 species, respectively. The remaining regions presented a richness of reptiles between 37 and 42 species. The results of the taxonomic distinctiveness showed that the SO region presented the highest value (31.56). The regions with the least taxonomic distinctiveness were SC and TD, with values of 25.14 and 27.92, respectively. For the snake group, all regions remained within the probability funnel,

indicating that the taxonomic distinctiveness of the snakes was consistent with what was expected (p > 0.05).

## Partitions of reptile beta diversity at the species and higher taxa level

At the species level, the total beta diversity of the reptiles was 75% (βSOR = 0.75), that of the lizards was 77% (βSOR = 0.77), and that of the snakes was 73% (βSOR = 0.74). In the three groups, turnover was the most important component, followed by differences in richness (reptiles: βSIM = 0.64, βSNE = 0.11; lizards: βSIM = 0.66, βSNE = 0.11; snakes: βSIM = 0.61, βSNE = 0.12; Fig. 3A).

In reptiles, the greatest dissimilarity occurred in five pairs of physiographic regions: PC/CP βsor = 0.76, CP/TD βsor = 0.758, PC/TD βsor = 0.755, TV/TD βsor = 0.74, and PC/SC βsor = 0.737. The replacement had the highest values for PC/CP βsim = 0.75, PC/SO βsim = 0.65, and PC/TV βsim = 0.64. Furthermore, 18 pairs of regions exhibited greater turnover than differences in richness. Only three pairs showed differences in richness greater than the turnover: TV/SC βsne = 0.44, SJ/SC βsne = 0.38 and SJ/TD βsne = 0.34 (Fig. 3B).

Beta diversity was greater in lizards than in reptiles, as six pairs of regions presented values greater than 70% dissimilarity (SO/TD βsor = 0.81, PC/TD βsor = 0.79, PC/CP βsor = 0.79, TV/TD βsor = 0.78, PC/TV βsor = 0.76, CP/TD βsor = 0.76). Regarding the turnover, 19 pairs presented higher values than the differences in richness, with the highest values occurring for PC/CP βsim = 0.77, PC/TV βsim = 0.73, PC/SO βsim = 0.64 and SJ/CP βsim = 0.62. Only two pairs of regions showed differences in richness greater than turnover (TV/SC βsne = 0.50, SJ/SC βsne = 0.39; Fig. 3C).

For the snakes, four pairs presented a total beta greater than 70% (CP/TD βsor = 0.75, PC/CP βsor = 0.73, PC/SC βsor = 0.72, TV/TD βsor = 0.71). Turnover prevailed over differences in richness in 17 pairs of regions, two of which had dissimilarities greater than 60% (PC/CP βsim = 0.73, PC/SO βsim = 0.65). Only three pairs presented differences in richness greater than turnover: TV/SC βsne = 0.41, SJ/SC βsne = 0.37, and SJ/TD βsne = 0.36 (Fig. 3D).

At the level of higher taxa, the beta diversity of reptiles was 66% (βSORT = 0.66), that of lizards was 64% (βSORT = 0.64), and that of snakes was 64% (βSORT = 0.64). Among the three groups analyzed, the turnover was greater than the difference in richness (reptiles: βSIMT = 0.46, βSNET = 0.19; lizards: βSIMT = 0.47, βSNET = 0.17; snakes: βSIMT = 0.46, βSNET = 0.18; Fig. 3E).

For reptiles, the highest taxonomic dissimilarity occurred in two pairs of physiographic regions with values greater than 60%: PC/TD βsorT = 0.62 and PC/SC βsorT = 0.62. The turnover was low since only three pairs presented turnover values higher than 35% (PC/CP βsimT = 0.42 and PC/TV βsimT = 0.39, PC/SO βsimT = 0.35). In addition, 11 pairs presented a greater turnover than differences in richness. Four pairs present higher values of differences in richness with values greater than 40% (TV/SC βsneT = 0.44, PC/TD βsneT = 0.43, PC/SC βsneT = 0.42, TV/TD βsneT = 0.40; Fig. 3F).

In lizards, the highest dissimilarity occurred in four pairs of regions, with a value greater than 55% (PC/SC: βsorT = 0.60, PC/TD: βsorT = 0.59, SO/SC: βsorT = 0.56, TV/TD:
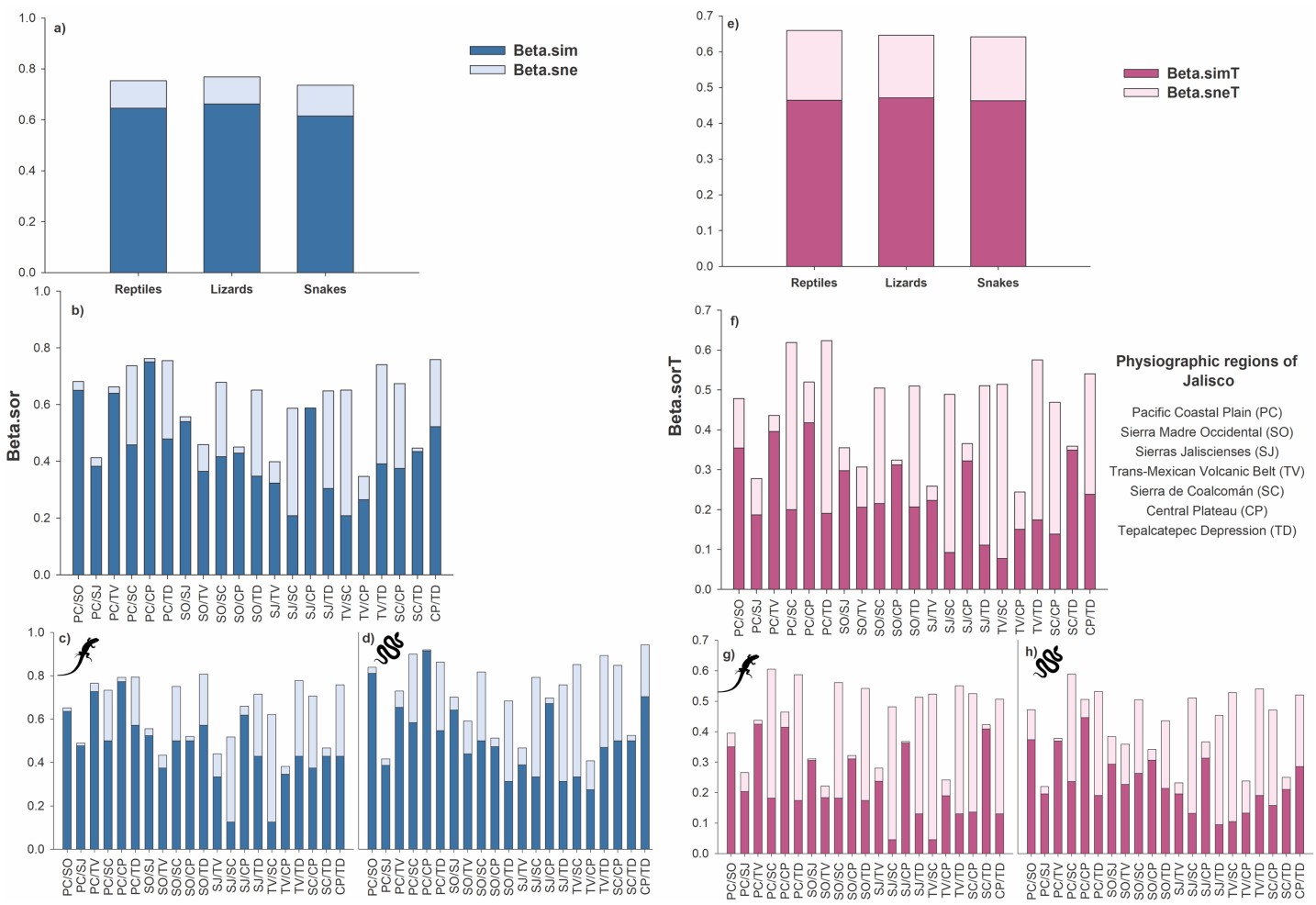

**Figure 3** **Total beta diversity for species and higher taxa of reptiles in the physiographic regions of the Jalisco state.** (A and E) Total beta diversity (beta.sor), replacement (beta.sim), and richness differences (beta.sne) for species and higher taxa of reptiles, lizards, and snakes in the entire set of physiographic regions of the state of Jalisco. (H and F) Total beta diversity, replacement, and richness differences for species and taxa of reptiles, (C and G) lizards, and (D and H) snakes for the all-possible combinations between pairs of the seven physiographic regions present in the Jalisco state. Lizard and snake silhouettes: https://images.app.goo.gl/wfvYcDTbBktp8mFZ7; snake cobra or anaconda silhouette icon long vector image (vectorstock.com).

βsorT = 0.55). The turnover was low, and the highest values were presented as follows: PC/TV: βsimT = 0.42, PC/CP: βsimT = 0.41, and SC/TD: βsimT = 0.41. Ten pairs present greater differences in richness, where five pairs presented values above 40% (TV/SC = βsneT = 0.48, SJ/SC: βsneT = 0.44, PC/SC: βsneT = 0.42, TV/TD: βsneT = 0.42, and PC/TD: βsneT = 0.41 (Fig. 3G)).

For snakes, the three pairs with the greatest dissimilarity were PC/SC: βsorT = 0.588, TV/TD: βsorT = 0.54, and PC/TD: βsorT = 0.531. Only one pair presented a value greater than 40% (PC/CP: βsimT = 0.44); however, 13 pairs exceeded the difference in richness, which was also low since only one pair exceeded 40% dissimilarity (TV/SC: βsneT = 0.42) and eight pairs presented values higher than the turnover value (Fig. 3H).

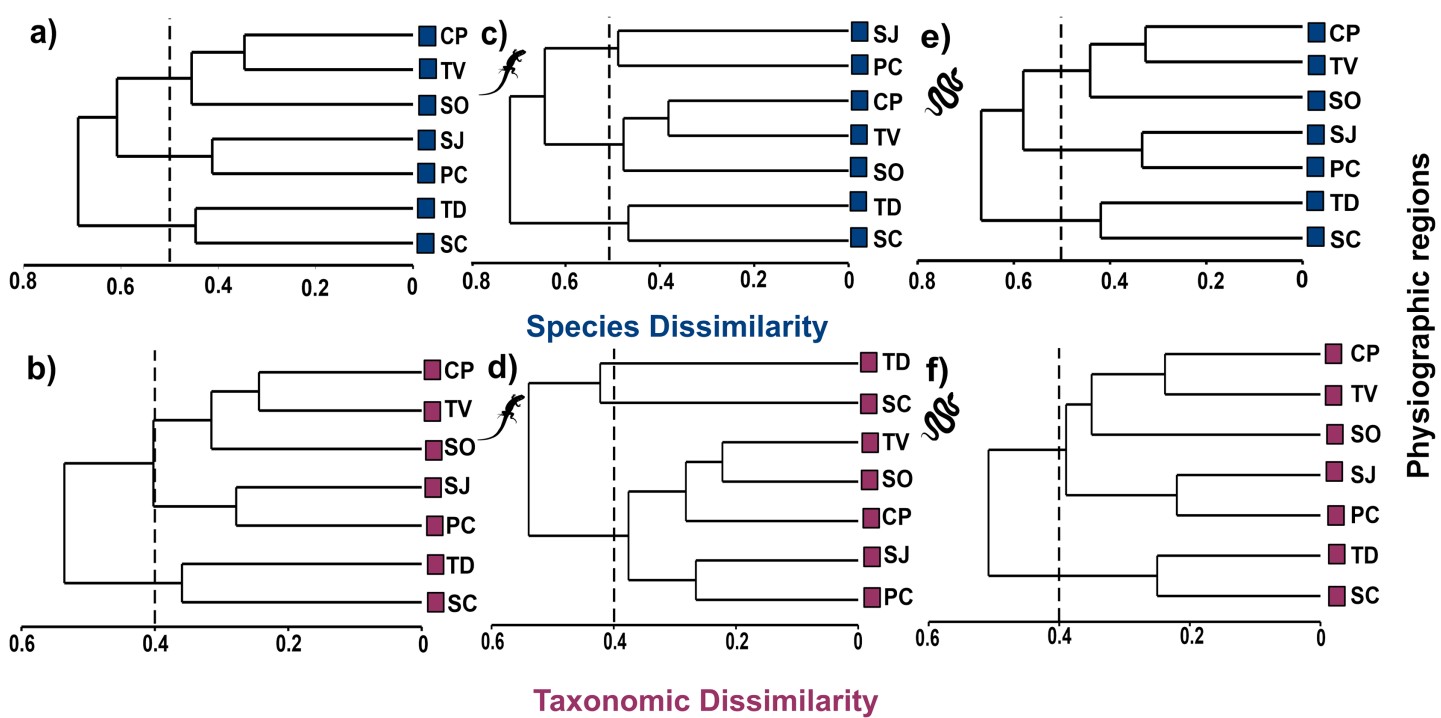

**Figure 4** Cluster analysis of species and taxonomic dissimilarity of reptiles (A and B), lizards (C and D) and snakes (E and F) between physiographic regions present in the Jalisco state. Lizard and snake silhouettes: https://images.app.goo.gl/wfvYcDTbBktp8mFZ7; snake cob.

### Dissimilarity of reptiles at the species and higher taxa level

Cluster analysis at the species level shows three groups at the 50% dissimilarity level, one formed by CP, TV, and SO ($\beta sor = 0.45$), the other by SJ and PC ($\beta sor = 0.41$), and the last one formed by TD and SC ($\beta sor = 0.45$, Fig. 4A). In higher taxa, the same groupings were observed as at the species level but with lower dissimilarity values, as observed in the CP, TV and SO ($\beta sorT = 0.32$), SJ and PC ($\beta sorT = 0.28$) and TD and SC groups ($\beta sorT = 0.36$; Fig. 4B).

The cluster analysis for lizards maintains the groupings of the set of reptiles, but with higher dissimilarity values. At the species level, the groups described above are maintained with dissimilarity values of $\beta sor = 0.49$ for SJ and PC; $\beta sor = 0.48$ for CP, TV and SO; and $\beta sor = 0.47$ for TD and SC (Fig. 4C). At the level of higher taxa, the groupings for lizards change, observing two isolated entities corresponding to TD and SC ($\beta sorT = 0.40$), and a single group formed by the remaining regions ($\beta sorT = 0.38$; Fig. 4D).

For snakes, the cluster analysis also maintains the groupings of the set of reptiles, but with the lowest values even compared to those of lizards. At the species level, the groups maintained dissimilarity values of $\beta sor = 0.33$ for SJ and PC; $\beta sor = 0.42$ for CP, TV and SO; and $\beta sor = 0.44$ for TD and SC (Fig. 4E). At the higher taxon level, only two groups are observed, one corresponding to CP, TV, SO, SJ, PC ($\beta sorT = 0.39$), and another formed by TD and SC ($\beta sorT = 0.25$; Fig. 4F).

### Relationships between beta diversity of reptile at the species and higher taxa level

The correlation analyses revealed a positive and significant relationship between the beta diversity of species and higher taxa for both total beta diversity (r = 0.88, $p < 0.0001$) and for its turnover (r = 0.83, p < 0.0001) and for differences in richness (r = 0.86, $p < 0.0001$). This relationship was also observed in the lizard ($\beta$sor: r = 0.77, $p < 0.0001$; $\beta$sim: r = 0.61, $p = 0.003$; $\beta$sne: r = 0.74, $p = 0.0001$) and snake ($\beta$sor: r = 0.93, $p < 0.0001$; $\beta$sim: r = 0.80, $p < 0.0001$; $\beta$sne: r = 0.85, $p < 0.0001$) groups (Table S8).

## DISCUSSION

In this study, we determined that the diversity of species and higher taxa of reptiles show different patterns between physiographic regions. We observed regions with high species richness but since they are more phylogenetically related, there is a lower representation of genera and families. Contrary to this, some regions show low species richness but with an overrepresentation of genera and families, which reflects a lower phylogenetic relationship between species. The differences between these two facets highlight the importance of quantifying community diversity by incorporating additional information on the evolutionary diversification of the taxa since a community composed of closely related species (*e.g.*, congeners) can be considered less diverse than a community with an equal number of species that are more distantly related (*Bevilacqua et al., 2021*).

The variety of higher taxa of the reptiles reported in this study through the taxonomic distinctiveness index was greater than that reported in other studies in which reptiles were also evaluated in different regions within and outside Jalisco (*Cruz-Elizalde et al., 2014*; *Cruz-Elizalde et al., 2022*; *Peña-Joya et al., 2018*). For example, *Cruz-Elizalde et al. (2014)* reported values around 65 of taxonomic distinctiveness, which were attributed to the presence of certain families of reptiles with a high number of species; for example, the Colubridae family in the snake group and Phrynosomatidae in the lizard group. On the other hand, *Cruz-Elizalde et al. (2022)* reported taxonomic distinctiveness values of less than 60, which was attributed to the presence of certain genera of reptiles with a high number of species, as well as a high degree of endemism. Compared to the above results, in our study, only the PC region had a greater value than those reported by these studies, indicating the presence of species less phylogenetically related, resulting in a greater variety of genera and families. The high taxonomic distinctiveness of PC for the reptiles and the group of lizards responds to the variety of ecosystems presented in this region, as well as the presence of tropical environmental conditions and its proximity to the sea, which allows it to be the only physiographic region where the presence of marine reptiles is recorded, highlighting the families Cheloniidae and Dermochelyidae; similarly, the Crocodylia order, with the sole representative *Crocodylus acutus*, inhabits estuarine ecosystems and mangrove vegetation (*Cruz-Sáenz et al., 2017*). These results coincide with those of *Maciel-Mata (2013)*, who noted that the areas with the greatest taxonomic diversity are characterized by being mainly in warm climates.

We determined that for the reptile group, the CP, SJ, and TV regions had a low variety of higher taxa, with taxonomic distinctiveness values similar to those reported by *Cruz-*

*Elizalde et al. (2014)* in the region of the Chihuahuan Desert. This low representativeness of taxa shows that some families and genera are overrepresented in these regions, such as the Phrynosomatidae family and particularly the *Sceloporus* genus, whose species are mostly distributed at high, medium and high elevations (*Cruz-Sáenz et al., 2017*), which are prevailing conditions in the regions CP, SJ and TV, respectively. In the case of snakes, the SO region stands out for having few species, but they are more distant taxonomically, as of the 63 species that are present, 21 belong to different genera and families, so this region is significant for the conservation of snakes and their evolutionary history (*Calderón-Patrón et al., 2016*). Furthermore, the genus *Micruroides* represented by *Micruroides euryxanthus* was only recorded for SO, which is closer to its main distribution (*Uriarte-Garzón et al., 2020*).

The beta diversity at the species level for multiple sites presented average values among the three groups of reptiles analyzed (75%), which exceeded the reptiles of Hidalgo (83%) of Isthmus of Tehuantepec in Oaxaca (82 and 79%, respectively; *Calderón-Patrón et al., 2013*) and those registered at the national level (*Koleff et al., 2008*; *Ochoa-Ochoa et al., 2014*; *Rodríguez et al., 2019*). In all the groups of reptiles analyzed, turnover prevailed over differences in richness, with values ranging between 61 and 64% dissimilarity. These values are similar to those presented for the reptiles in Hidalgo (62% replacement; *Calderón-Patrón et al., 2016*) and for those obtained by *Villegas-Patraca et al. (2022)* for reptiles in Baja California (0.46% to 0.78%). However, in the case of the reptiles of the Isthmus of Tehuantepec, turnover and differences in richness contribute in very similar proportions (*Calderón-Patrón et al., 2013*).

For total beta diversity between pairs of regions for all reptiles, the highest values were presented for five pairs (PC/CP, CP/TD, PC/TD, TV/TD, and PC/SC). The most contrasting dissimilarity values were presented between PC and SC, geographically distant regions with notable environmental differences. These results coincide with those recorded for Hidalgo and the Isthmus of Tehuantepec in Oaxaca, where the areas with greater distances and environmental differences are those that present the greatest dissimilarity in their species compositions of reptiles due to replacement (*Calderón-Patrón et al., 2013*; *Calderón-Patrón et al., 2016*).

The total beta diversity, turnover, and differences in richness of reptiles at the level of higher taxa were lower in all the cases than at the species level because as we move up in the taxonomic categories, the dissimilarity decreases drastically, as each category increasingly includes more species and/or taxa (*Calderón-Patrón et al., 2016*). This is consistent with that reported by *Qian (2009)*, who mentions that the ratio of beta diversity from species to genus level is 1.24, and to family level 1.85; regarding the ratio of beta diversity from genus to family level is 1.50. On the other hand, despite the differences in dissimilarity values, our correlation results show that the patterns between beta diversity at the species level and higher taxa are similar, including the turnover and differences in richness components for the three groups of reptiles analyzed. This result indicates that the beta diversity assessments of higher taxa are good surrogates for analyzing beta diversity at the species level (*Calderón-Patrón et al., 2016*).

The dendrograms show that, in general, the physiographic regions are grouped similarly for all reptiles. The observed groups may respond to the geographical distance between the regions and their greater environmental similarity, such as CP, TV, and SW, which are contiguous and share tropical deciduous forests, xerophilous scrubs, and pine and oak forests. SJ and PC both have dry forests, while CP and TV share tropical deciduous forests and pine and oak forests. These results coincide with clusters of ecoregions in Hidalgo, that share some types of vegetation and have a similar composition of terrestrial vertebrates (*Calderón-Patrón et al., 2016*).

Finally, we determined that overall beta diversity in lizards and snakes was similar, contrary to what we expected. In lizards, the species show a distribution restricted to one or two regions possibly due to endemism and its habitat requirements; for example, *Iguana iguana*, endemic to Mexico, is distributed in tropical and subtropical forests with habitat preference near bodies of water, so they only register in the PC region (*Gómez-Mora, Suazo-Ortuño & Alvarado-Díaz, 2012*). On the other hand, *Phrynosoma asio* is a native species with distribution from northern Nayarit along the Pacific coast of Mexico to at least the Isthmus of Tehuantepec (*Köhler, 2021*), which only records for PC. Species of the genus *Sceloporus* also exhibit restricted distributions in the state, for example, *Sceloporus spinosus* (endemic to Mexico), inhabits coniferous forests and xerophilous scrublands with terrestrial, saxicolous, and occasionally arboreal habits (*Díaz de la Vega-Pérez et al., 2022*), it was reported only in the CP region. Although to a lesser extent, snakes also follow this restricted distribution, mainly in the genera *Crotalus, Geophis*, and *Tantilla*. *Crotalus*, for example, has the greatest richness and endemism on the continent, where its species have specialized in different habitats and microhabitats, so many of its species have restricted distributions (*Paredes-García, Ramírez-Bautista & Martínez-Morales, 2011*). The genera *Geophis* and *Tantilla* have a high number of endemisms, fossorial habits, and specialized microhabitats that explain their restricted distribution (*Canseco-Márquez et al., 2016; Wilson & Mata-Silva, 2014*). Another genus that contributes to the replacement of four species distributed in a single region is *Thamnophis*, which due to its aquatic habits its distribution depends on the presence of bodies of water (*Rossman, Ford & Seigel, 1996*).

## CONCLUSIONS

In this study, we provided some answers to the questions presented at the outset. Our findings show that alpha diversity of species and higher taxa maintain different patterns, indicating that Jalisco presents regions with overrepresentation of reptile families and genera, as well as regions with an opposite trend. The representation of higher taxa is higher in the lizard group, although in terms of species richness snakes are the most prominent group. Beta diversity shows similar patterns at the species and higher taxa level, even in the turnover components and differences in richness. We determined that turnover is the most important component of beta diversity at both the species and higher taxa levels, for all reptile groups. These findings show that incorporating phylogenetic information about species through taxonomic relationships provides complementary information that species diversity *per se*, especially at the level of alpha diversity.

# ACKNOWLEDGEMENTS

The authors are grateful to all reviewers for their comments and suggestions to improve this manuscript.

### Funding

This research was supported by project 272002 of the Programa de Fortalecimiento de la Investigación y el Posgrado [Research and Postgraduate Strengthening Program] of the Universidad de Guadalajara awarded to the Laboratorio de Ecología, Paisaje y Sociedad [Ecology, Landscape and Society Laboratory], CUCOSTA-UdeG; and by project 270478 of the Programa de apoyo a la mejora en las condiciones de producción de los miembros del SNI y SNCA [Support program for the improvement in production conditions of members of the SNI and SNCA] of the Universidad de Guadalajara awarded to Karen Elizabeth Peña-Joya, Jorge Téllez-López and Eréndira Canales-Gómez. The funders had no role in study design, data collection and analysis, decision to publish, or preparation of the manuscript.

### Grant Disclosures

The following grant information was disclosed by the authors:
Universidad de Guadalajara: 272002, 270478.

### Competing Interests

The authors declare that they have no competing interests.

### Author Contributions

- Jaime Manuel Calderón-Patrón conceived and designed the experiments, performed the experiments, analyzed the data, prepared figures and/or tables, authored or reviewed drafts of the article, and approved the final draft.
- Jorge Téllez-López conceived and designed the experiments, prepared figures and/or tables, authored or reviewed drafts of the article, and approved the final draft.
- Eréndira Canales-Gómez analyzed the data, prepared figures and/or tables, authored or reviewed drafts of the article, and approved the final draft.
- Karen Elizabeth Peña-Joya conceived and designed the experiments, performed the experiments, analyzed the data, prepared figures and/or tables, authored or reviewed drafts of the article, and approved the final draft.

### Data Availability

The results from beta diversity partitions for species and higher taxa of lizards, snakes, and reptiles between pairs of physiographic regions are available in the Supplemental Files.

## Supplemental Information

Supplemental information for this article can be found online at http://dx.doi.org/10.7717/peerj.18343#supplemental-information.

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
