# Peer review of "Ecological and taxonomic dissimilarity in species and higher taxa of reptiles in western Mexico"

_PeerJ, doi:10.7717/peerj.18343_

## Round 0.1 · original submission · Major Revisions

Thank you very much for your manuscript titled “Ecological and taxonomic dissimilarity in species and higher taxa of reptiles in western Mexico” that you sent to PeerJ.

This study presents very valuable and relevant information on regional patterns of reptile diversity in part of Mexico, comparing diversity patterns at the local and regional level.

As you will see below, comments from referee 1 suggest a minor revision while reviewers 2 and 3 suggest a major revision before your paper can be published. Given this, I would like to see a major revision dealing with the comments. Their comments should provide a clear idea for you to review, hopefully improving the clarity and rigor of the presentation of your work. I will be happy to accept your article pending further revisions, detailed by the referees, which largely focus on separate the analysis by groups of reptiles to understand possible patterns of distribution, richness and diversity. In addition to considering the distribution of the species or genera to interpret the results obtained. Also, some statistical points require more clarity. In discussion consider the ecology of the species and groups.

Reviewer 1 suggests in highlighting the distribution of the species and focusing the analysis by groups and focusing part of the discussion on the ecology of the species. He also makes some notable observations about the manuscript that need to be addressed.

Reviewer 2's observations are noted directly on the manuscript, basically focusing on restructuring the manuscript.

Reviewer 3 emphasizes defining the research question and including its answer within the conclusions, in addition to reviewing and reformulating the description of the results of the cluster analysis.

Please note that we consider these revisions to be important and your revised manuscript will likely need to be revised again.

·

Basic reporting

The study in general has a good structure, the analysis is adequate according to the amount of information. The authors manage the methods and interpretation of the results, in addition to knowing the biological group.

Experimental design

The objective of the study, as well as the methodology and sources of information, are adequate. In general, the source of information is from a publication by other authors, so it is not possible to control the effort or original methodology used; however, for the analyzes used it is not necessary to know it. Likewise, it highlights the way to separate the analysis into reptiles, lizards and snakes, since different results can be observed.

Validity of the findings

The results obtained are interesting considering the scale of work, and can be used to compare at the regional, landscape or country level. However, little is described about distribution patterns by groups. It is not a biogeography study of groups of reptiles, but it is necessary to know the distribution of the species or genera to interpret the results obtained, beyond highlighting the regions with higher or lower diversity values ​​and beta values.

Additional comments

In general, the manuscript presents very important information due to the scale of work, as well as the region that has generally been little studied. Separating the analysis by groups, even though they are reptiles, helps understand patterns of distribution, richness and diversity of reptiles, and this can be compared at a regional level. In general, there is also a lot of discussion that there are over- or under-represented groups of taxa, but nothing is mentioned about the ecology of the species and these groups. For example, if the values ​​in lizards are greater in dissimilarity by region, why not describe which groups give these values ​​by province? This should not be a problem since the species are the same as those in the publication where the richness of reptiles for the state of Jalisco is described.

I strongly suggest that you consider mentioning the genera or families according to their distribution patterns and even displacement, since it is these qualities that shape the associations of reptile communities at the local scale, and from there, at the regional scale.

Reviewer 2 ·

Basic reporting

Needs improvement. Please see the Reviewer’s comment PDF for details.

Experimental design

Experimental design is well. But statistical analysis needs revision/improvement.

Validity of the findings

Nice.

Annotated reviews are not available for download in order to protect the identity of reviewers who chose to remain anonymous.

Reviewer 3 ·

Basic reporting

In this study, the authors assess the ecological and taxonomic distinctiveness of reptiles through the partitioning of beta into two dimensions of biodiversity (species diversity and taxonomic diversity). The analysis is based on biogeographic subprovinces and the pattern of Beta diversity for each dimension is explained for reptiles in general and for lizards and snakes.
The English language is clear; however, it is suggested to revise some sections of the Introduction and Results where information is not clear. (L53-58) Clarify how is the pattern of species replacement at different scales. What does it mean for communities that dissimilarity occurs by replacement or from differences in richness. The introduction (43-47) should include appropriate literature to contextualize the area of knowledge. Therefore, it is necessary to include citations that relate to the environmental crisis on a global and regional level.
There is no research question guiding the study. Therefore, it should be incorporated and supported with adequate background and theories.
One of the weaknesses of the work is the carelessness in presenting the results in two sections. Therefore, it is necessary to reconsider the description of the clusters, as it is not clear how the described grouping is done for each dissimilarity dendrogram presented in the sections "Cluster analysis of species dissimilarity (lines 252-267)" and "Clsuter analysis of taxa with greater dissimilarity (269 - 282)".
Some minor notes in the text are:
64 Indicate which wildlife group you are referring to.
103-104 Use Spanish or English to refer to the names of the subprovinces, but do not mix them.
107 revise the citation in Valero Padilla (with hyphen or just Valero)
109 the seven physiographic regions of Jalisco be treated in English or Spanish?
192 should indicate that this is the group of snakes
196 the value of 98.70 does not match the values shown in Figure 2c.
Use replacement instead of Exchange in all sections.

Experimental design

Although the authors point out the lack of studies in this area and in particular beta diversity analyzes for the reptile group by biogeographic subprovinces. They do not provide a clearly defined research question or the research problem to be addressed. It is therefore suggested to justify the relevance of the work in this context. The indices used for beta analysis are adequate and current. The methods are described in sufficient detail and contain information to reproduce them.

Validity of the findings

The authors provide additional and sufficient data to understand both the results and the context of the study area.
Conclusions should be drawn in relation to the research question underlying the study. And clarify the conservation implications of the dissimilarity between the provinces compared, as well as differences due to species replacement and differences in richness.

Additional comments

It is important to revise and reformulate the description of the results of the cluster analysis and, if necessary, the corresponding discussion. The labels of the figures should be revised, and the form and criteria used for grouping the subprovinces should be clarified.

---

## Round 0.2 · accepted · Accept

After reviewing this revised version of your manuscript, I see that the main comments suggested by the reviewers have been included. Therefore, I am satisfied with the current version and consider it ready for publication.

·

Basic reporting

The study is important overall. It is a contribution that analyzes diversity patterns at a regional scale, and that makes use of complete sources of information.

The references are sufficient with respect to the methodology, introduction and discussion, and the general layout of the manuscript is punctual. The objectives of the study are met according to the data and analysis used.

Experimental design

Experimental design, methods and analysis appropriate to the study and objectives set out.

Validity of the findings

Conclusions appropriate and consistent with the objectives, methodology and discussion.

Reviewer 2 ·

Basic reporting

I read the revised manuscript “Ecological and taxonomic dissimilarity in species and higher taxa of reptiles in western Mexico”. It is glad to see that the authors addressed all the issues/concerns raised and revised the manuscript accordingly. I recommend accepting this manuscript for publication in PeerJ.

Experimental design

Accurate.

Validity of the findings

Acceptable.